# Patterns in the temporal complexity of global chlorophyll concentration

Vitul Agarwal [1] ✉, Jonathan Chávez-Casillas [2], Keisuke Inomura [1] & Colleen B. Mouw [1]

Decades of research have relied on satellite-based estimates of chlorophyll-*a* concentration to identify oceanographic processes and plan in situ observational campaigns; however, the patterns of intrinsic temporal variation in chlorophyll-*a* concentration have not been investigated on a global scale. Here we develop a metric to quantify time series complexity (i.e., a measure of the ups and downs of sequential observations) in chlorophyll-*a* concentration and show that seemingly disparate regions (e.g., Atlantic vs Indian, equatorial vs subtropical) in the global ocean can be inherently similar. These patterns can be linked to the regularity of chlorophyll-*a* concentration change and the likelihood of anomalous events within the satellite record. Despite distinct spatial changes in decadal chlorophyll-*a* concentration, changes in time series complexity have been relatively consistent. This work provides different metrics for monitoring the global ocean and suggests that the complexity of chlorophyll-*a* time series can be independent of its magnitude.

Satellites have been used to monitor global ocean color for decades, and satellite-derived products have been critical in determining the global trends of primary productivity[1–3], coastal runoff[4,5], sea ice extent[6,7], and harmful algal blooms[8]. Of particular relevance is chlorophyll-*a* concentration ([chl-*a*]), the primary pigment used by phytoplankton to perform photosynthesis, which can be reliably estimated from the reflectance of blue and green light from the oceans[9]. Chlorophyll-*a* concentration estimates are used for various goals: estimating primary productivity[3], developing ecological indicators[10], monitoring long-term trends[11], or testing earth-system-models[12,13]. Despite the necessity of using [chl-*a*] time series in global studies, it is unclear what role, if any, the time series complexity of [chl-*a*] time series plays in accounting for region-specific differences in global ocean color.

The roughness or complexity of a time series can strongly indicate multiple phenomena: high stochasticity in a target process, measurement error, long tails in the data, rapidly changing system states, or the confluence of all these factors and more. As phytoplankton populations display chaotic dynamics, non-linear behavior, and intermittent instability[14–16], measurements of the complexity of [chl-*a*] time series might also capture large-scale patterns that structure global phytoplankton communities. There

are many ways of estimating the natural complexity of a time series, such as calculating the fractal or Hausdorff dimension[17], permutation entropy[18], or Lyapunov exponents[19,20]. These are all different measures of complexity for dynamic systems and can often be related to one another. Although some of these calculations have been performed at select locations[21], a global analysis of [chl-*a*] time series complexity would provide a deeper understanding of the spatio-temporal characteristics of phytoplankton blooms as observed by satellite radiometers.

We used approximately 25 years (1998-2022) of global [chl-*a*] observations that were based on measurements made with the Sea-viewing Wide Field of View Sensor (SeaWiFS), Moderate Resolution Imaging Spectroradiometer on the Aqua satellite (MODIS), Medium Resolution Imaging Spectrometer (MERIS), Visible and Infrared Imaging/Radiometer Suite (VIIRS) and the Ocean and Land Color Instruments (OLCI-A and OLCI-B) sensors. Chlorophyll concentration estimates were from the Garver-Siegel-Maritorena Model merged data product[22,23] at 25 km resolution and daily temporal resolution, available from the Hermes GlobColour website (https://hermes.acri.fr/). Using a merged product allowed us to increase the spatial and temporal coverage of our analysis; however, we also tested our analysis on

[1]Graduate School of Oceanography, University of Rhode Island, Narragansett, RI, USA. [2]Department of Mathematics and Applied Mathematical Sciences, University of Rhode Island, Kingston, RI, USA. ✉e-mail: vitulagarwal@uri.edu

single missions (i.e., MODIS) and found only small differences on a global scale (Figure S1). Previous studies have used this product to understand phytoplankton bloom phenology based on change point algorithms[24,25], cycles in chlorophyll dynamics[26], the climate-driven components of [chl-$a$] variability[27], and the contributions of citizen science programs[28].

Here, we show how two different complexity metrics can indicate region-specific differences in several phenomena, such as the regularity of daily changes in [chl-$a$] or the likelihood of extreme values within a [chl-$a$] time series. Some of these differences are likely driven by differential rates of satellite spatial coverage, ocean color algorithm error, and naturally occurring dynamics. On a decadal time scale, we observe changes in both the mean magnitude of [chl-$a$] and the complexity properties of the time series. This suggests that our understanding of the global ocean may need to account for intrinsic variation within [chl-$a$] time series, as well as how we perceive and estimate [chl-$a$] from satellite radiometers.

## Results and Discussion

An important component in global [chl-$a$] studies is the definition of a phytoplankton bloom, often described as an anomalous or distinct increase in the biomass of a particular species (or set of species). Determining when a bloom occurs is not trivial, with varying strategies that can produce different results[29,30]. One example is defining the bloom start date as the point where [chl-$a$] rises a certain percentage above the median [chl-$a$][31–33]. We defined an analogous term to what economists call elasticity to analyze the sensitivity of [chl-$a$] time series to thresholds, where the elasticity of a [chl-$a$] time series is a measure of the responsiveness of the percentage change between two variables. The variables measured were the number of times that there was an increase in daily [chl-$a$] above a certain threshold and the threshold itself (see Methods for further details). Higher elasticity describes a rougher time series and a greater sensitivity to thresholds, likely due to greater variability within day-to-day [chl-$a$] change (Fig. 1). As elasticity does not heavily weigh the magnitude of [chl-$a$], but instead the variability in the magnitude of [chl-$a$], it is not a direct measure of

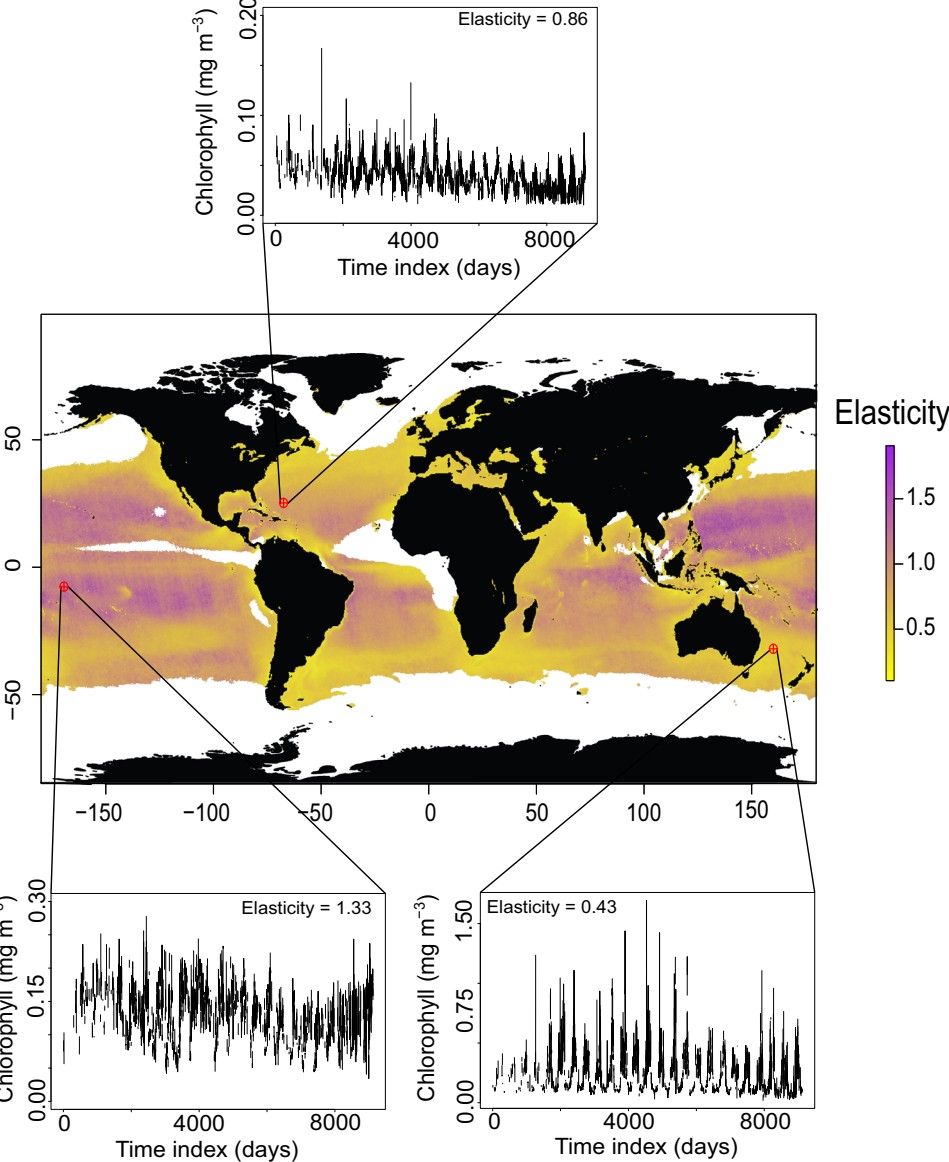

**Fig. 1 | Global map of elasticity calculated from 25 years (1998-2022) of daily-scale chlorophyll-$a$ concentration time series.** Darker colors (purple) indicate greater elasticity for the corresponding 25x25km pixel, whereas lighter colors (yellow) indicate lower elasticity. Some examples of time series with their elasticity values are provided. Areas in white are pixels where greater than 80% of the time series were missing observations or less than 400 days with consecutive measurements. 'Time index' refers to the day of sampling starting from January 1st, 1998, to January 1st, 2023.

phytoplankton growth dynamics or productivity. It captures the regularity of change within a time series for each pixel, irrespective of the mechanisms involved in driving those changes.

When viewed globally, the elasticity of [chl-*a*] time series varied across broad ocean regions. The oligotrophic gyres, known for their low chlorophyll concentration and primary production rates[34,35], had the highest elasticity values. In contrast, regions with high chlorophyll concentration and prominent seasonal cycles, such as the Patagonian Shelf, the Baltic Sea, the west coast of Central America, and the southwest Pacific Ocean[36–39], showed lower elasticity values. As elasticity is sensitive to the regularity of [chl-*a*] change, regions with strong, periodic drivers of phytoplankton blooms (often manifested in seasonal or annual patterns) would naturally have lower values of elasticity in comparison to regions where the periodicity of [chl-*a*] change is irregular. Figure S2 highlights how elasticity can vary for three different model time series. If all daily changes within a time series were equal, its elasticity would equal to 0 (no sensitivity to thresholds). When evaluated for the relative frequency of daily changes, some regions were prominent in their dynamic behavior, with more days of significant [chl-*a*] increase regardless of any chosen threshold (Figure S3). Interestingly, the North Atlantic gyre had lower than average elasticity values than both Pacific gyres, despite similar values of [chl-*a*]. The difference is likely due to the stronger seasonal patterns in the North Atlantic and the influence of the annual spring bloom[40,41], which would impart more regularity in the [chl-*a*] time series in the North Atlantic. Our results suggest that the elasticity metric tracks the relative strength of climatic and seasonal forcing of [chl-*a*] across broad ocean regions and might provide a quantitative measure of differentiating between them. Global Earth-system models may benefit by matching the relative strength of climatic drivers to the measured elasticity of [chl-*a*] time series on a global scale. For observational studies that rely on measurements of [chl-*a*], more dynamic behavior of time series in different ocean regions might necessitate an increase in the precision and number of samples taken to identify trends.

Another metric often used to compute the complexity of a time series is the fractal dimension. When considering a time series, we can think of it as the graph of function from $\mathbb{R} \to \mathbb{R}$ where $\mathbb{R}$ refers to the set of real numbers. As a graph, it then becomes a subset of $\mathbb{R}^2$. It is known that when the function is smooth (i.e., differentiable), its fractal dimension is 1, but when this function is not differentiable, the fractal dimension might be greater. This dimension cannot be larger than 2 as the graph is an embedding of $\mathbb{R}^2$. The fractal dimension is an indicator of how rough a curve is. It can also be understood as the autocorrelation of the time series as the lag period tends to 0. A higher fractal dimension implies higher instantaneous autocorrelation of the time series. For example, it is known that the fractal dimension of a Brownian motion, a non-differentiable continuous function, is 1.5. In this sense, the fractal dimension gives us a way to quantify how volatile a time series and how likely we are to observe extreme values on it.

When we computed the fractal dimension of every [chl-*a*] pixel time series, the values across the global ocean were remarkably consistent (typically above 1.85), with some key differences in particular regions (Fig. 2). The subtropical North and South Pacific, tropical North Atlantic, the Amazon plume region, and the Eastern coast of Madagascar had lower-than-average fractal dimensions. Part of the reason for this difference could be tied to the influence of physical ocean dynamics: such as the South Equatorial current and freshwater discharge from the Amazon River[42,43], eddy formation from the Agulhas current near Madagascar[44–46], the Gulf Stream and hurricane formation in the tropical North Atlantic[47,48] and the influence of tropical instability waves in the equatorial Pacific[49]. Another reason could be the episodic nature of elevated [chl-*a*] in many of these regions, which may lead to a greater number of outliers within the [chl-*a*] time series. As the box-counting method has been shown to return lower fractal dimension values in time series with sharp changes[50], we first remove most outliers from each time series (see Methods) before calculating the fractal dimension. This was done to minimize the error in our computations due to likely erroneous measurements. In this case, our global map of fractal dimension reveals the likelihood of anomalous spikes within each time series (long tails in the data), possibly due to naturally occurring dynamics, but also, atmospheric correction and ocean color algorithm error. Regions less prone to outliers have time series that return higher fractal dimensions due to fewer anomalous spikes.

We conducted several additional analyses to further test the sensitivity of elasticity and fractal dimension to our methodological choices. As the estimation of satellite chlorophyll using merged data can often contain bias due to the differences among individual sensors[51–54], we calculated elasticity and fractal dimension for MODIS data alone (2002-2022) and compared it against the merged dataset for the entire time period. The results were largely similar on a global scale (Figure S1), with some minor differences in the magnitude of elasticity for the oligotrophic gyres. Similarly, when we tested for the influence of time series resolution (daily, weekly, monthly, etc.) on the metrics, our results showed that aggregated time series are typically smoother (i.e., possess lower fractal dimension). The elasticity also changes with coarser resolution for some regions and indicates geographically similar, but greater differences (Figure S4). To get an estimate of deviation for each complexity metric, we re-calculated the elasticity and fractal dimension for every time series based on random selected windows of 4000 days each. The standard error based on 30 trials suggests that elasticity can vary up to ±0.20 in some regions such as the oligotrophic ocean, whereas fractal dimensions only vary up to ±0.04. Figure S5 highlights the deviation of both complexity metrics for the global dataset. Lastly, we also compared both complexity metrics to some traditional measures of variability for each time series (mean [chl-*a*], standard deviation, relative standard deviation, and average seasonal amplitude). The calculated fractal dimension of each time series showed a nonlinear positive relationship, whereas the elasticity showed a nonlinear negative relationship with each metric (Figures S6-7). When evaluated against each other on a similar scale, elasticity and fractal dimension did not present any strong relationships (Figures S8-9).

When calculated annually, both the elasticity and the average fractal dimension of the global ocean had a significant shift in 2003 and a gradual return to 1998 levels thereafter (Fig. 3a–c), whereas the mean [chl-*a*] showed a general decrease till 2019. Interestingly, a second transition around 2019 shows a rise in mean [chl-*a*], a sharp increase in fractal dimension and a drop in elasticity. The error was larger prior to 2003 for every metric, likely due to fewer observations and the sole data source being SeaWiFS. The satellite data record from 2003 onwards contained merged observations and included data from MODIS-Aqua and MERIS. As data collection by SeaWiFS stopped in 2010 and by MERIS in 2012, while data collection by VIIRS began in 2012, it is possible that the sharp changes around the years 2010-2012 can be attributed to changes in the temporal and spatial coverage of the time series during this period. Similarly, although previous studies have noted little to no inter-annual differences in [chl-*a*] over the 2003 transition[26], changes in mean complexity over this period are likely tied to the increase in spatial coverage due to a greater number of sensors in orbit. Even though we observed a decline in [chl-*a*] since 2003 in agreement with the existing literature[55,56], the true estimates of changes in [chl-*a*] likely depend on the methods used to determine trends and the composition of the spatial dataset[57,58]. We only considered part of the global ocean based on data availability (i.e., fewer high-latitude regions). Consequently, the mean [chl-*a*] only provides context for the changes in elasticity and fractal dimension. Our results suggest a decline in global [chl-*a*] and a concurrent change in the structure of [chl-*a*] time series in the satellite record.

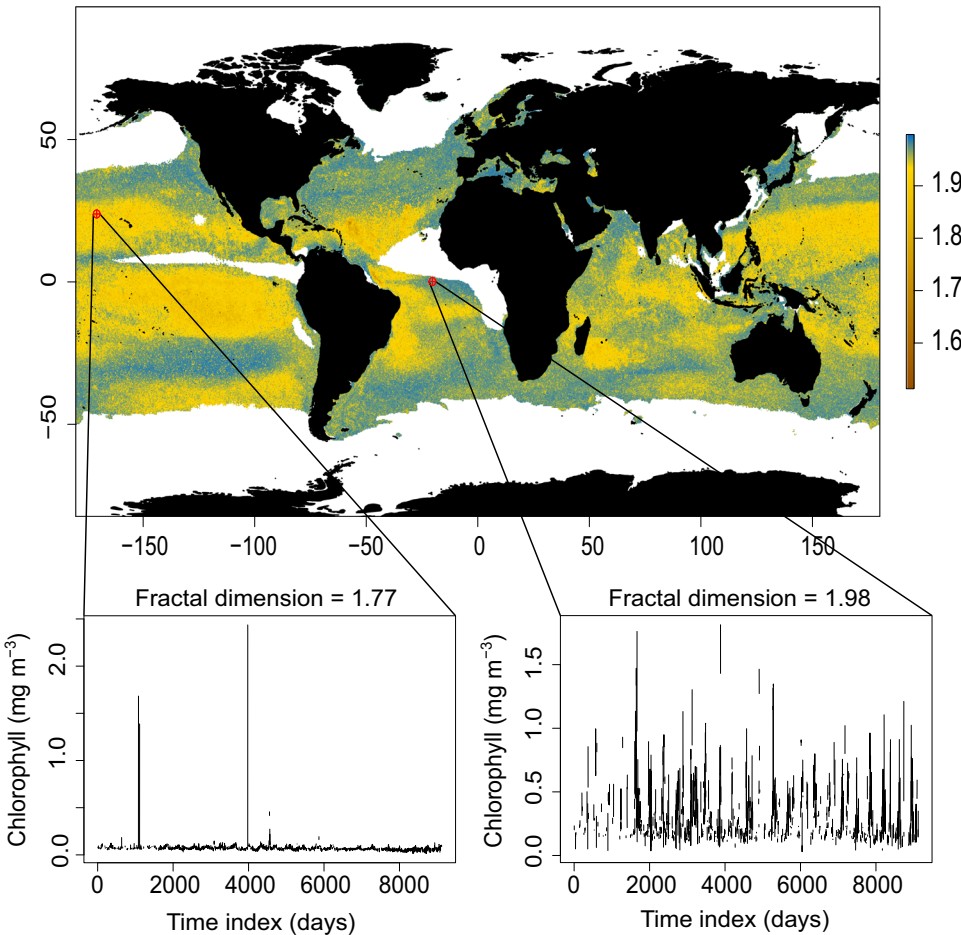

**Fig. 2 | Global map of fractal dimension calculated from 25 years (1998–2022) of daily-scale chlorophyll-*a* concentration time series with random resampling of any missing observations.** Blue and yellow indicate a higher fractal dimension for the corresponding 25x25km pixel, whereas brown indicates lower fractal dimensions. Most of the global ocean has relatively high fractal dimensions (global mean = 1.95). Some examples of time series with their fractal dimension values are provided. Areas in white are pixels where greater than 80% of the time series were missing observations. 'Time index' refers to the day of sampling starting from January 1st, 1998, to January 1st, 2023. Note: the color scale is not linear and has been chosen to highlight spatial differences.

These results have several implications. First, an increase in the fractal dimension and a decrease in elasticity imply a lower frequency of anomalous events in many regions and an increase in stronger, repeatable cycles of [chl-*a*]. Part of this change can likely be attributed to the greater number of satellite sensors in orbit (and merged in the [chl-*a*] product) and a change in the magnitude of error-related outliers. Errors may decrease due to the greater availability of data and possibility of quality control but may also increase over time as the hardware of individual satellites age. Further improvements in satellite capabilities may allow future studies to increase the precision of their conclusions and enhance the predictability of [chl-*a*] time series by quantifying more accurately the impact of stochasticity and the measurement error[59]. While we did not evaluate the chlorophyll-*a* product uncertainty in this study, future studies may consider accounting for the difference between [chl-*a*] algorithms, different uncertainty calculations, and their relationship to time series complexity[60–62]. One possibility is that temporal complexity is driven by high variability within certain regions of the visible spectrum; for example, slight day-to-day differences in the green spectrum may propagate across [chl-*a*] calculations and lead to greater roughness within the time series. This variability could be caused by several factors – such as rapid differences in the taxonomic composition of phytoplankton or the influence of terrestrial discharge.

Second, when evaluated for differences across space, we also found that changes in mean [chl-*a*] and both the complexity metrics from 2003-2022 were inconsistent for different ocean regions (Fig. 3d–f). For example, areas such as the Black Sea, parts of the Western equatorial Pacific, and the subtropical Indian Ocean had an overall increase in [chl-*a*] in 2022 compared to 2003. In contrast, changes in the elasticity and fractal dimension were more spatially consistent across the global ocean. These patterns indicate that both metrics are insensitive to changes in [chl-*a*] magnitude and, thus, independent of the processes that control [chl-*a*] magnitude in different regions.

Third, although there is a long history and multiple benefits to categorizing marine ecological provinces based on environmental, geophysical, and biogeochemical data sources[63–66], measurements of time series complexity offer a valuable addition to monitoring global ocean change on a larger scale. Our results could form the basis of further refinement in the definitions of marine ecological provinces. Future studies may also consider how disparate regions in the global ocean are similar in time series complexity and whether the patterns of observations by satellite radiometers are tied to specific environmental or ecological drivers of [chl-*a*]. Examples of possible drivers include mixed layer stability, eddy formation, temperature change and plankton growth and grazing rates.

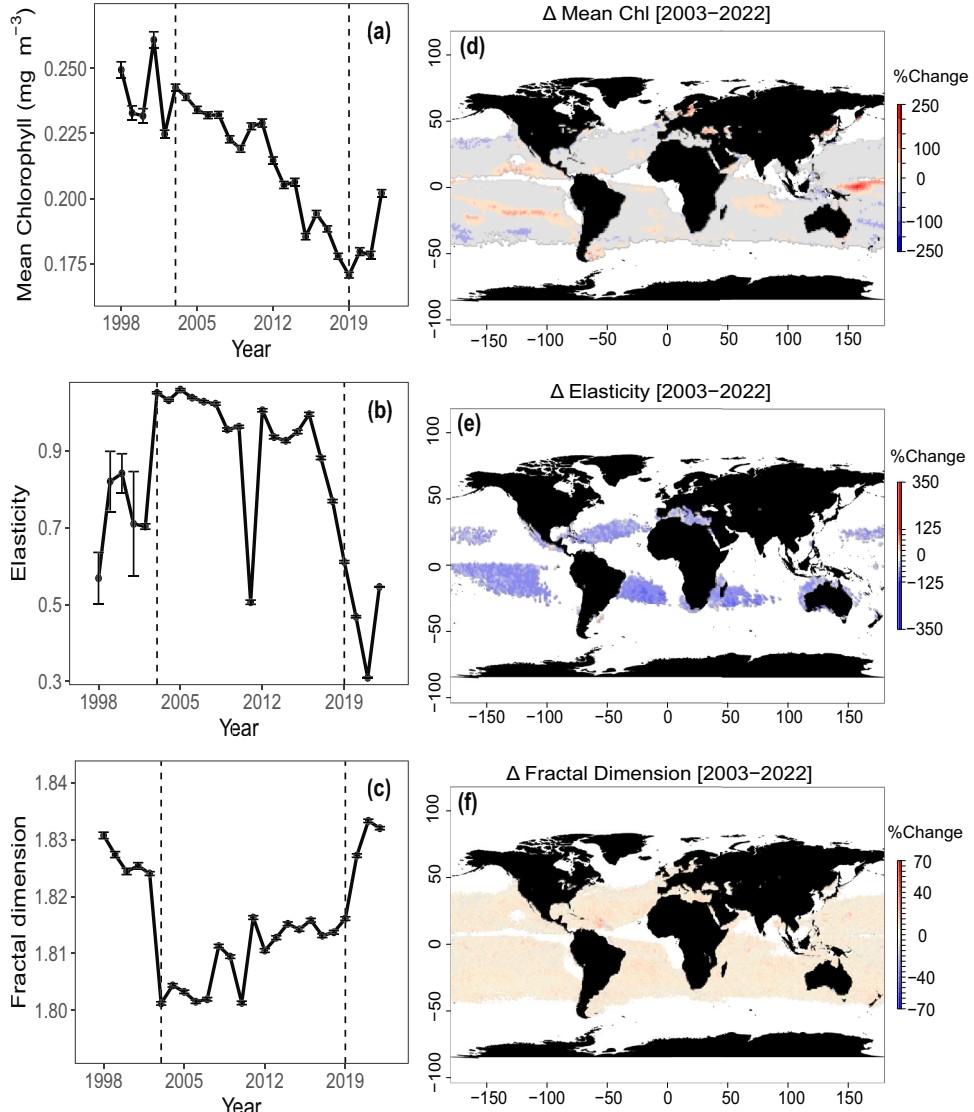

**Fig. 3 | Inter-annual trends in time series complexity.** (**a**) mean chlorophyll-a concentration (mg m⁻³), (**b**) mean elasticity and (**c**) mean fractal dimension calculated yearly (1998–2022) from daily-scale time series data. Black points indicate the mean value and error bars are the 95% confidence intervals ($1.96 \times S.E.$). The sample sizes for each point can be found in Supplementary Table 1. Gridlines mark the years of 2003 and 2019. Global maps of (**d**) the percentage change in mean chlorophyll-*a* concentration between the years 2003 and 2022, (**e**) the percentage change in mean elasticity between the years 2003 and 2022 and (**f**) the percentage change in mean fractal dimension between the years 2003 and 2022. Blue values indicate negative change, whereas red values indicate positive change. White pixels are the locations where greater than 80% of the entire time series were missing observations, or where there were less than 100 consecutive days of observation in 2003 and 2022 each. Each pixel is 25x25km. There are more white pixels in subplot (**e**) because of less than 100 days of consecutive samples in 2003. We need more instances of repeated daily coverage to calculate elasticity (see Methods).

## Methods

We generated a 25-year daily time series (1998-2022) of [chl-*a*] by extracting a merged data product at 25 km resolution from https://hermes.acri.fr/. The merged data product used the Garver-Siegel-Maritorena Model[22,23], which combined data from the Sea-viewing Wide Field of View Sensor (SeaWiFS), Moderate Resolution Imaging Spectroradiometer on the Aqua satellite (MODIS), Medium Resolution Imaging Spectrometer (MERIS), Visible and Infrared Imaging/Radiometer Suite (VIIRS) and the Ocean and Land Color Instruments (OLCI-A and OLCI-B) sensors.

Every time series on a daily scale contains some gaps due to insufficient satellite coverage and quality flags that were removed (Figure S10). To proceed with the analysis and remove any blank pixels, we instituted a cut-off of 20% to filter through the entire global time series dataset. Only the time series with greater than 20% of days

sampled over 25 years were considered in further analysis. This ensured a minimum of ~1800 days had available [chl-*a*] estimates for every time series. These thresholds were chosen to strike a balance between data availability and spatial coverage. In our analysis, we wanted to maximize global spatial coverage without compromising on our ability to accurately estimate complexity due to sampling gaps.

We used a multi-step process to calculate the elasticity of every pixel time series. First, we created a 1-day-lagged time series and subtracted it from the original time series, giving us estimates of daily [chl-*a*] change. Any missing samples in the original time series were carried forward to the time series of daily change. Only the time series that had at least 400 samples of daily [chl-*a*] change proceeded to the next step in the calculation. Next, we created two uniform distributions of thresholds ($\tau_1$ and $\tau_2$; 1000 values each) that ranged from 1% to 50% of the median of the individual [chl-*a*] time series. For every time series,

we then calculated the number of days where daily [chl-$a$] change exceeded the threshold ($X_1$ and $X_2$). This calculation occurred independently for both sets of thresholds. Third, we calculated the relative difference in the threshold ($\varepsilon$) used to find $X_1$ and $X_2$ according to Eq. 1, and the relative difference in daily change ($N$) between $X_1$ and $X_2$ according to Eq. 2.

$$\varepsilon = \frac{\tau_2}{\tau_1} - 1 \qquad (1)$$

$$N = \frac{X_2}{X_1} - 1 \qquad (2)$$

Lastly, a linear model was created to find the slope between $\log_e \varepsilon$ and $\log_e N$ according to Eq. 3. The absolute value of the slope ($\beta_1$) of this linear model was termed as the elasticity of the time series (as there are no non-negative values). The greater the value of $\beta_1$, the stronger the negative relationship between $\log_e \varepsilon$ and $\log_e N$.

$$\log_e N = -\beta_1 \log_e \varepsilon + \beta_2 \qquad (3)$$

It is known from the classical theory of linear models[67], that the slope coefficient of a linear regression has the interpretation of being the average amount by which the response changes when the predictor changes by a unit. However, when considering a regression like the one in Eq. 3, where both predictor and response variable are measured in logarithmic scale, the coefficient of the predictor can be computed as

$$\frac{\partial \log_e N}{\partial \log_e \varepsilon} = \frac{\left(\frac{\partial N}{N}\right)}{\left(\frac{\partial \varepsilon}{\varepsilon}\right)} \qquad (4)$$

which corresponds to what the economists denote as *elasticity*. Some examples of model time series and their respective elasticity values can be found in Figure S2 in the Supplementary material. As elasticity is based on relative change and is not sensitive to the magnitude of specific points, we did not remove any outliers from any time series for this part of the analysis. We also did not attempt to measure statistical significance, nor remove any pixels based on significance thresholds. Future studies may consider detailed statistical analyses to quantify the false discovery rate, the chosen number of threshold combinations, or the applicability of linear assumptions in the estimation of elasticity.

For the second part of the analysis, we used the R package fractaldim[68] to calculate the fractal dimension of every [chl-$a$] time series. First, we cleaned every time series by removing outliers that exceeded 3 standard deviations from the mean [chl-$a$] value. This was done to remove any aberrant spikes of [chl-$a$] that were likely incorrect and were missed in quality control. On average, there were about 0.004% of all pixel time series that were removed as outliers (~1–2 days every year across all locations). Given the large number of sampling gaps in some cases, we also performed bootstrapping for every pixel time series, which is a resampling technique. To be precise, the gaps in each time series were filled in by randomly selecting values (with replacement) from within the measurements of the same time series. This method of resampling was chosen to ensure we had a complete time series for every pixel without adding bias. We then calculated the fractal dimension of the time series using the box-counting method. The number of missing values and the range of [chl-$a$] (max - min) for every pixel time series did not show any relationship with the calculated fractal dimension (Figure S11). Examples of model time series and their calculated fractal dimension can be found in Figure S12 in the Supplementary material.

For calculating the yearly elasticity and fractal dimension, the process was similar to the one described above, except the time series

length was limited to the sampling points in each year from 1998-2022. To ensure enough data, we only calculated the fractal dimension and the mean [chl-$a$] of the time series (before random resampling) if at least 100 days were sampled for that year. Similarly, we only calculated the elasticity if there were at least 100 consecutive samples for that year. The results were reported in Fig. 3 as numerical averages for every pixel with error bars representing the 95% confidence intervals ($1.96 \times S.E.$). Table T1 in the supplementary material reports the number of samples used to calculate the annual [chl-$a$] mean, fractal dimension, and elasticity. Percent change for either metric (represented as $\rho$) was calculated according to Eq. 5:

$$\% \, \text{change} = \frac{\rho_{2022} - \rho_{2003}}{\rho_{2003}} \times 100 \qquad (5)$$

All the analyses and plotting were conducted in R[69] using the R packages "raster"[70], "maps"[71], "ggplot2"[72], "cowplot"[73], "fractaldim"[68], "MASS"[74], "viridis"[75] and "ncdf4"[76].

## Reporting summary
Further information on research design is available in the Nature Portfolio Reporting Summary linked to this article.

## Data availability
Remote sensing data are available from https://hermes.acri.fr/. Data can be downloaded for specific locations and time periods under "GlobColour data search". Sample datasets (processed) to reproduce part of the analysis can be found at https://doi.org/10.5281/zenodo.10498362[77]. Source data are provided with this paper.

## Code availability
The code required for the analysis is available at https://github.com/vitul-agarwal-1/chl-complexity (https://doi.org/10.5281/zenodo.10498357). All the analyses and plotting were conducted in R[69] using the R packages raster[70], maps[71], ggplot2[72], cowplot[73], fractaldim[68], MASS[74], viridis[75] and ncdf4[76].

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

## Acknowledgements

VA and CBM acknowledge support from the Rhode Island Sea Grant (Grant number: NA22OAR4170123 to CBM). CBM acknowledges support from CINAR (Cooperative Institute for the North Atlantic Region; Grant number: NA19OAR4320074 to CBM). This work was supported by a grant from the Simons Foundation (LS-ECIAMEE-00001549 to KI). We would like to acknowledge the team at the Unity high-performance computing cluster located at the Massachusetts Green High Performance Computing Center (MGHPCC).

## Author contributions

VA conducted the analysis and wrote the first draft of the manuscript. JC, KI and CBM assisted with all aspects of this work.

## Competing interests

The authors declare no competing interests.
