## [Peer Review File · Nature Communications]

Patterns in the temporal complexity of global chlorophyll concentrationREVIEWER COMMENTS

Reviewer #1 (Remarks to the Author):

The authors analyzed features (elasticity and fractal dimension) of satellite chlorophyll time series at global scale which would be useful to assess changes in spatial and time variability patterns of the ocean productivity. The authors introduced a novel viewpoint in the analysis of these chlorophyll time series which permitted detecting significant regional differences as well as conspicuous interannual changes. I think that the research might be useful for improving our understanding of the drivers dealing to shifts in the ocean productivity that probably are linked to the global change. However, I have several doubts about the robustness of the results:

(1) My main concern is that the satellite chlorophyll time series were built by using data from different platforms (sensors) which operate during a given time period. Consequently, the source of satellite data is not temporally or (probably) spatially homogenous. Published literature shows that the estimation of satellite chlorophyll trends using time series built with merged data from different sensors must be performed with caution (Morozov et al., 2010; Sathyendranath et al., 2017; Staehr et al., 2022). For instance, Gómez-Jakobsen et al. (2022) observed systematic biases between trends calculated with VIIRS and MODIS for the same time period. The same authors found that SeaWiFS tends to overestimate chlorophyll concentration compared with MODIS. These biases were attributed to differences in the performance of the sensors that might imply differences in response or processing of the reflectance signal. In any case, these findings imply that the chlorophyll concentration calculated from different platforms might be affected by technical factors which are difficult to assess. Even the performance of each sensor would vary following the latitudinal gradient and/or for areas optically dissimilar (for instance, coastal vs. open sea areas). I guess that the impact of these biases among sensors should be lesser if aggregated data are used (for instance, monthly or yearly means) or in areas where reflectance signal is more stable (open sea compared to coastal waters). However, in the case of the presented work, elasticity was calculated from daily increases in the chlorophyll a data series that 'a priori' would be highly sensitive to these biases. In some way, the authors recognize this limitation in their analyses as in lines 166-168 is explained that some changes observed would be due to

differences among sensors. I think that the authors can easily discard if their results are significantly affected by multi-platform signal usage. For instance, elasticity and fractal dimension might be calculated independently for time series obtained from each single satellite for overlapping periods. The comparison of these parameters will aid to analyze the robustness of the methodology beyond the technical limitations of the satellites.

(2) The elasticity calculation is based on the phytoplankton bloom concept which is fine. However, normally a bloom is defined not only by their intensity but also by its duration (blooms last several days, at least). I am also in agreement with occurrence of punctual daily increases of chlorophyll beyond the thresholds for a given location would be a good descriptor of the time series but I am not sure that it can be directly related to phytoplankton growth dynamics. In addition of the technical inconsistencies mentioned by the authors that can produce outliers (and that were not eliminated for the calculation of elasticity unless I am mistaken), sudden changes in surface chlorophyll for a given location can be due to water advection from other nearby areas including discharges from terrestrial streams in coastal water. I think that these other sources that produce instabilities in the chlorophyll time series should be filtered to get a calculation of elasticity more clearly related to phytoplankton productivity. Perhaps, introducing some criteria to select the number of days where daily chlorophyll change exceeded the thresholds based on the duration of the peak would be useful.

(3) The authors showed that different areas had values of elasticity and/or fractal dimensions some different. For instance, in Figure 2, the two areas had fractal dimensions of 1.77 and 1.98. Furthermore, these values are provided without some deviation measurement. The authors suggest that these differences are really relevant meaning that is indicative of dissimilar phytoplankton dynamics. However, both figures only differ by approx. 10%. Consequently, for the reader, the relevance of the magnitude of these differences would be not evident. I think that the authors should use some statistical approach to assess if differences in elasticity and/or fractal dimension between two areas or time series should be considered significant.

Other specific comments

(4) Page 2, line 31. The concept of complexity of time series is not unambiguous. I will expect some definition is provided in these lines.

(5) Page 2, line 37. The expression “some aspects” sound vague. Please, specify which aspects of the ocean may be changing.

(6) Page 2, line 48. I am not sure that “light emitted” is the most suitable term in this context. Perhaps, reflectance is more appropriate.

(7) Page 3, line 52. As commented above, the concept “complexity” has to be explained.

(8) Page 6, lines 106-108. I cannot fully understand why stronger annual cycles should give lower elasticity values (I guess that stronger annual cycles mean higher amplitude of the seasonal cycle). Additionally, it is not clear what the authors mean with “annual cycles” as differenced from seasonal cycles. On the other side, this sentence contradicts “The difference is likely due to the stronger seasonal patterns in the North Atlantic” since the North Atlantic gyre had higher elasticity values.

(9) Page 7. Line 138. I guess that these outliers that are attributed to technical inconsistencies or measurement errors also affected the calculation of the elasticity. However, it appears that they were not removed in the time series when this parameter was calculated. Please, justify.

(10) Page 12. Line 228. I wonder why exactly these thresholds were chosen.

(11) Page 13, line 238. Since elasticity was calculated from the linear model, I wonder if the statistical significance of the model was estimated for each time series. I suppose that the elasticity value calculated for a given time series should be considered no representative if the fitting of the data to the linear model was not significant.

(12) Page 14, line 260. Please, explain why this technique to fill gaps was used.

Figures 1 and 2. It is unclear what is the “time index” represented in the x-axis.

References

- Gómez-Jakobsen, F., Ferrera, I., Yebr, L., Mercedo, J.M., 2022. Two decades of satellite surface chlorophyll a concentration (1998–2019) in the Spanish Mediterranean marine waters (Western Mediterranean Sea): Trends, phenology and eutrophication assessment. *Rem. Sensing Appl. Soc. Environ.* 28, 100855. <https://doi.org/10.1016/j.rsase.2022.100855>
- Morozov, E., Korosov, A., Pozdnyakov, D., Pettersson, L., Sychev, V., 2010. A new area-specific bio-optical algorithm for the Bay of Biscay and assessment of its potential for SeaWiFS and MODIS/Aqua data merging. *Int. J. Rem. Sens.* 31, 6541–6565. <https://doi.org/10.1080/01431161.2010.508802>.
- Staehr, S.U., Van der Zande, D., Staehr, P.A.U., Markager, S., 2022. Suitability of multisensory satellites for long-term chlorophyll assessment in coastal waters: a case study in optically-complex waters of the temperate region. *Ecol. Indicat.* 134, 108479 <https://doi.org/10.1016/j.ecolind.2021.108479>.
- Sathyendranath, S., Brewin, R.J.W., Jackson, T., M'elin, F., Platt, T., 2017. Ocean-colour products for climate-change studies: what are their ideal characteristics? *Remote Sens. Environ.* 203, 125–138. <https://doi.org/10.1016/j.rse.2017.04.017>

Reviewer #2 (Remarks to the Author):

Overview

The authors have prepared an interesting manuscript detailing the application of two statistical methods previously unused for the analysis of ocean chlorophyll. These metrics will likely be of use in future research works. The finding of potential global similarities in these metrics is also of interest. However as it stands the manuscript needs some additional work as much of the interpretation does not appear to be supported by the results; I have provided some suggestions where initial analysis may be employed below. I look forward to seeing the manuscript following review.

Major comments

The main area that should be targeted for improvement is in the interpretation of the novel application of these statistics. This particularly applies to what the statistics are revealing, as the current interpretation does not have the analysis to support it and is at times conflicting. The manuscript states these metrics measure time series complexity, however what these reflect in the real world, be it measurement error, blooms, seasonal cycling, climatic forcing or day to day variability needs to be supported by some additional analysis. I am also curious how these new metrics compare to standard deviation / variance and should be better contrasted against each other.

- Conflicting interpretations:
 - o L107 implies low elasticity is linked to stronger seasonal cycles
 - o L110-112 implies high elasticity is linked to stronger seasonal cycles
 - o Figure S3 implies elasticity is insensitive to regular seasonal cycles
- Suggestions for possible additional analysis:
 - o Application of statistics to data with seasonality removed
 - o Comparison with global seasonal amplitude
 - o Comparison with global variance (/standard deviation)
 - o Comparison with chlorophyll product uncertainty
- ♣ This is available with the GlobColour product as provided by CMEMS although I'm not sure if it is temporally varying

In summary sufficient additional analysis needs to be performed to (a) compare with traditional measures of variability and to (b) support the interpretation of the physical mechanism behind changes in the values of the statistics.

Additionally further attempts should be made to highlight similarities and differences in the two statistical measures discussed. For example the spatial patterns in figures 1 and 2 look very similar, could these be shown on the same colour map with same standardised scale, to better highlight the similarities / differences (at least as supplementary)? Additionally it would be more useful if the examples of each of figure 1 and 2 covered the same grid points, including examples where the two metrics agree and also where they don't.

Finally Figures 3 (e) and (f) have vastly different colour scales with broad ranges, which may appear to show global homogeneity but may instead be an artefact of the plotting, these should be redrawn with the full range of colours for each being employed (at least as supplementary)

Specific / minor comments

L112-114 I don't feel this statement is supported by the results as they stand, indeed no mention of climatic forcing has been made up to this point.

L34 / L139 If the goal of the research is to analyse outliers, outliers should probably not have been removed, as stated in L141. Alternatively a distinction needs to be made between the outliers in these different contexts.

L161 - Some comments should be made on the change in the time series after 2019.

177-179 - Errors may increase over time as the hardware of individual satellites age.

Figures 3 a-c – Gridlines may help identify years where changes are occurring

Figure 3f - Why does this subplot have more white pixels than the subplots above it?

Figure 3 – The order of elasticity and fractal dimension should ideally follow the text (i.e. the plots of elasticity should appear above those of fractal dimension)

L228 - Day to day thresholds in the range of 0.01 to 0.5 % of median seem low, and I would expect changes to cross this threshold the vast majority of days, are these the correct values and if so, could the authors comment on this?

Reviewer #3 (Remarks to the Author):

The authors proposed new metric to quantify the complexity of Chl time series obtained from satellite ocean color remote sensing, and implied that the patterns could be explained using seasonal forcing and outlier events. This reviewer found the analyses interesting, but the results and implications are questionable.

First, the authors used daily satellite Chl products to carry out the analyses, but it is well known that daily satellite Chl is subject to many sources of uncertainties, thus the spikes in the time series (see Figs. 1 and 2) are not necessarily real oceanographic events. Note that such spikes are the main drivers of the "time series complexity" studied here. To the least,

the authors should test sensitivity of the new metric on different temporal resolution and spatial resolution.

More importantly, for a specific location, the metric provides a measure of Chl variation in the past decades only, has no information on the future change or variation of Chl, or if there will be a spike on any days or periods. While complexity of the time series is useful information, it is more important to know, and understand, the temporal variation of Chl (or any other physical and biogeochemical properties) in the global ocean.

The reviewer comments are in blue, and our replies are in black. The page and line numbers we cite refer to the **track changes version of the manuscript**.

Reviewer Comments & Suggestions:

Reviewer #1 (Remarks to the Author):

The authors analyzed features (elasticity and fractal dimension) of satellite chlorophyll time series at global scale which would be useful to assess changes in spatial and time variability patterns of the ocean productivity. The authors introduced a novel viewpoint in the analysis of these chlorophyll time series which permitted detecting significant regional differences as well as conspicuous interannual changes. I think that the research might be useful for improving our understanding of the drivers dealing to shifts in the ocean productivity that probably are linked to the global change. However, I have several doubts about the robustness of the results:

Thank you for your thorough review. Following the reviewer's comments, we have added several analyses and hope that the new manuscript inspires greater confidence in the results.

(1) My main concern is that the satellite chlorophyll time series were built by using data from different platforms (sensors) which operate during a given time period. Consequently, the source of satellite data is not temporally or (probably) spatially homogenous. Published literature shows that the estimation of satellite chlorophyll a trends using time series built with merged data from different sensors must be performed with caution (Morozov et al., 2010; Sathyendranath et al., 2017; Staehr et al., 2022). For instance, Gómez-Jakobsen et al. (2022) observed systematic biases between trends calculated with VIIRS and MODIS for the same time period. The same authors found that SeaWiFS tends to overestimate chlorophyll a concentration compared with MODIS. These biases were attributed to differences in the performance of the sensors that might imply differences in response or processing of the reflectance signal. In any case, these findings imply that the chlorophyll concentration calculated from different platforms might be affected by technical factors which are difficult to assess. Even the performance of each sensor would vary following the latitudinal gradient and/or for areas optically dissimilar (for instance, coastal vs. open sea areas). I guess that the impact of these biases among sensors should be lesser if aggregated data are used (for instance, monthly or yearly means) or in areas where reflectance signal is more stable (open sea compared to coastal waters). However, in the case of the presented work, elasticity was calculated from daily increases in the chlorophyll a data series that 'a priori' would be highly sensitive to these biases. In some way, the authors recognize this limitation in their analyses as in lines 166-168 is explained that some changes observed would be due to differences among sensors. I think that the authors can easily discard if their results are significantly affected by multi-platform signal usage. For instance, elasticity and fractal dimension might be calculated independently for time series obtained from each single satellite for overlapping periods. The comparison of these parameters will aid to analyze the robustness of the methodology beyond the technical limitations of the satellites.

We understand and largely agree with the reviewer's concern. Our decision to use a merged product allowed us to increase our spatial coverage of the global oceans and reduce gaps within each time series. Picking individual sensors would have allowed us to avoid issues of data merging but would have simultaneously exacerbated issues of incomplete temporal and spatial coverage.

Regardless, we followed the reviewer's suggestion and performed additional analyses for a specific sensor – MODIS (Figure A below). The results for the entire merged dataset (Figures 1 and 2 in the manuscript) and the results for MODIS data are largely similar on a global scale for both observed spatial patterns and actual measurement of roughness. These results suggest that our global analysis is not driven by “multi-platform signal usage”.

We agree with the reviewer that there may be differences due to the performance of a sensor, but these likely matter for smaller regional analyses. Future studies may need to exhaustively evaluate region-specific differences in time series complexity for different sensors across a range of different time periods.

In the revised manuscript, we have added some text and supplementary figures highlighting the MODIS-only analysis.

Ln 80-82: “Using a merged product allowed us to increase the spatial and temporal coverage of our analysis; however, we also tested our analysis on single missions (i.e. MODIS) and found only small differences on a global scale (Figure S1).”

Ln 175-180: “As the estimation of satellite chlorophyll using merged data can often contain bias due to the differences among individual sensors (51–54), we calculated elasticity and fractal dimension for MODIS data alone (2002-2022) and compared it against the merged dataset for the entire time period. The results were largely similar on a global scale (Figure S1), with some minor differences in the magnitude of elasticity for the oligotrophic gyres.”

Figure A: Global map of elasticity (top) and fractal dimension (bottom) calculated from 20-years (2002-2022) of daily-scale chlorophyll-*a* concentration time series as derived from MODIS observations alone. Darker colors (purple) indicate greater elasticity for the corresponding 25x25km pixel, whereas lighter colors (yellow) indicate lower elasticity. Similarly, Green and yellow indicates a higher fractal dimension for the corresponding 25x25km pixel, whereas red indicates lower fractal dimensions. Please note that the color scale for the map of fractal dimensions was chosen to highlight spatial differences. Areas in white are pixels where greater than 80% of the time series were missing observations or less than 400 days with consecutive measurements.

(2) The elasticity calculation is based on the phytoplankton bloom concept which is fine. However, normally a bloom is defined not only by their intensity but also by its duration (blooms last several days, at least). I am also in agreement with occurrence of punctual daily increases of chlorophyll beyond the thresholds for a given location would be a good descriptor of the time series but I am not sure that it can be directly related to phytoplankton growth dynamics. In addition of the technical inconsistencies mentioned by the authors that can produce outliers (and that were not eliminated for the calculation of elasticity unless I am mistaken), sudden changes in surface chlorophyll for a given location can be due to water advection from other nearby areas including discharges from terrestrial streams in coastal water. I think that these other sources that produce instabilities in the chlorophyll time series should be filtered to get a calculation of elasticity more clearly related to phytoplankton productivity. Perhaps, introducing some criteria to select the number of days where daily chlorophyll change exceeded the thresholds based on the duration of the peak would be useful.

The reviewer is correct in noting that the complexity of a time series implicitly includes information on phytoplankton growth dynamics, water advection, terrestrial discharge etc. As our goal was to measure the complexity of different time series, our analysis was conducted to avoid being entirely driven by phytoplankton productivity. Our use of two different metrics captures some of the different aspects of complexity inherent in each time series.

Elasticity, the way it is currently designed, does not account for the duration of any bloom. The primary goal was to make an estimate of roughness in the intensity of chlorophyll. It is for the same reason that the magnitude of chlorophyll concentration does not feature prominently in the elasticity metric, but rather the *variability* in magnitude that drives the calculation. The fractal dimension, on the other hand, likely does include information on ‘bloom duration’, as it estimates roughness using the box-counting method.

There are several challenges that prevent the inclusion of bloom duration into the elasticity metric.

- (1) gaps in the data – As there are gaps within every time series, any estimates of bloom duration require detailed gap-filling. This is a particularly difficult problem to solve, as bloom duration is a time-sensitive quantity, which means the order of points matter in the calculation. This automatically excludes resampling as an appropriate strategy to fill such gaps. Furthermore, other techniques such as interpolation would alter the elasticity of the time series and overestimate the influence of bloom duration in roughness.
- (2) non-intuitive metric - the measurements of intensity and duration are separate, and this means that we would need to create a multi-parameter metric that can simultaneously capture the dynamics of both. Such a convoluted measure might not represent actual phytoplankton productivity and would be harder to interpret for most readers.

We have now added some text to our paper describing how the elasticity metric does not represent phytoplankton growth dynamics.

Ln 97-100: “As elasticity does not heavily weigh the magnitude of [chl-a], but instead the variability in the magnitude of [chl-a], it is not a direct measure of phytoplankton growth dynamics or productivity. It captures the regularity of change within a time series for each pixel, irrespective of the mechanisms involved in driving those changes.”

(3) The authors showed that different areas had values of elasticity and/or fractal dimensions some different. For instance, in Figure 2, the two areas had fractal dimensions of 1.77 and 1.98. Furthermore, these values are provided without some deviation measurement. The authors suggest that these differences are really relevant meaning that is indicative of dissimilar phytoplankton dynamics. However, both figures only differ by approx. 10%. Consequently, for the reader, the relevance of the magnitude of these differences would be not evident. I think that the authors should use some statistical approach to assess if differences in elasticity and/or fractal dimension between two areas or time series should be considered significant.

Thank you for this valuable comment. In the revised manuscript, we have added an additional supplemental figure and associated discussion to help illustrate the deviation of each metric for each location.

This was done using jack-knifing, i.e. performing the calculations on randomly assigned 4000-day windows within each time series (x30) to create a distribution of values with associated variance. The estimated deviation for each time series is fairly low and does not change the interpretation of our results.

Ln 183-194: “To get an estimate of deviation for each complexity metric, we re-calculated the elasticity and fractal dimension for every time series based on random selected windows of 4000-days each. The standard error based on 30 trials suggests that elasticity can vary up to ± 0.20 in some regions such as the oligotrophic ocean, whereas fractal dimensions only vary up to ± 0.04 . Figure S5 highlights the deviation of both complexity metrics for the global dataset.”

Figure B: Global map of the standard error in elasticity (top) and fractal dimension (bottom) calculated from 25-years (1998-2022) of daily-scale chlorophyll-*a* concentration time series. Darker colors (red) indicate greater error for the corresponding 25x25km pixel, whereas lighter colors (yellow) indicate lower error. Areas in white are pixels where greater than 80% of the time series were missing observations or less than 400 days with consecutive measurements. Error was calculated from a distribution of elasticity and fractal dimension values for each pixel (30 randomly selected temporal windows of 4000-days each).

Other specific comments

(4) Page 2, line 31. The concept of complexity of time series is not unambiguous. I will expect some definition is provided in these lines.

Thank you, we added an explanation for what we meant by time series complexity.

Ln 31-34: “Here we develop a metric to quantify time series complexity (i.e. a measure of the ups and downs of sequential observations) in chlorophyll-*a* concentration and show that seemingly disparate regions (e.g. Atlantic vs Indian, equatorial vs subtropical) in the global ocean can be intrinsically similar.”

(5) Page 2, line 37. The expression “some aspects” sound vague. Please, specify which aspects of the ocean may be changing.

We changed “some aspects” to “the temporal complexity”.

Ln 37-38: “This work offers novel metrics for monitoring the global ocean and suggests that the temporal complexity of the ocean may be changing as a whole, rather than regionally.”

(6) Page 2, line 48. I am not sure that “light emitted” is the most suitable term in this context. Perhaps, reflectance is more appropriate.

We adopt the reviewer’s suggestion.

Ln 47-49: “Of particular relevance is chlorophyll-*a* concentration ([chl-*a*]), the primary pigment used by phytoplankton to perform photosynthesis, which can be reliably estimated from the reflectance of blue and green light from the oceans”

(7) Page 3, line 52. As commented above, the concept “complexity” has to be explained.

We addressed this comment above.

(8) Page 6, lines 106-108. I cannot fully understand why stronger annual cycles should give lower elasticity values (I guess that stronger annual cycles mean higher amplitude of the seasonal cycle). Additionally, it is not clear what the authors mean with “annual cycles” as differenced from seasonal cycles. On the other side, this sentence contradicts “The difference is likely due to the stronger seasonal patterns in the North Atlantic” since the North Atlantic gyre had higher elasticity values.

We apologize for the confusion. We have added additional text to explain what we mean. In brief, elasticity measures the roughness of a time series by testing its sensitivity to thresholds. This means that areas that tend to have regular cycles of change would have lower elasticity values as compared to areas with irregular change. This can be seen in the supplemental figure S1. If every change in a time series was equal, the elasticity would be 0.

For the contradictory statement, we apologize for the error. The sentence should read “lower than average” instead of “higher than average”.

Ln 123-134: “As elasticity is sensitive to the regularity of [chl-*a*] change, regions with strong, periodic drivers of phytoplankton blooms (often manifested in seasonal or annual patterns) would naturally have lower values of elasticity in comparison to regions where the periodicity of [chl-*a*] change is irregular. Figure S2 highlights how elasticity can vary for three different model time series. If all daily changes within a time series were equal, its elasticity would equal to 0 (no sensitivity to thresholds). When evaluated for the relative frequency of daily changes, some regions were prominent in their dynamic behavior, with more days of significant [chl-*a*] increase regardless of any chosen threshold (Figure S3). Interestingly, the North Atlantic gyre had lower than average elasticity values than both Pacific gyres, despite similar values of [chl-*a*]. The difference is likely due to the stronger seasonal patterns in the North Atlantic and the influence of the annual spring bloom (40, 41), which would impart more regularity in the [chl] time series in the North Atlantic.”

(9) Page 7. Line 138. I guess that these outliers that are attributed to technical inconsistencies or measurement errors also affected the calculation of the elasticity. However, it appears that they were not removed in the time series when this parameter was calculated. Please, justify.

We did not remove any outliers when calculating elasticity as the metric is not sensitive to the magnitude of specific points – it is based on the percentage of change based on different thresholds. For any anomalous outliers, there would be no effect to the slope of the linear model. We have added a statement in our Methods section.

Ln 345-346: “As elasticity is based on relative change and is not sensitive to the magnitude of specific points, we did not remove any outliers from any time series for this part of the analysis.”

(10) Page 12. Line 228. I wonder why exactly these thresholds were chosen.

These thresholds were chosen to create a balance between data availability and spatial coverage. We add a statement mentioning this.

Ln 309-311: “These thresholds were chosen to strike a balance between data availability and spatial coverage. In our analysis, we wanted to maximize global spatial coverage without compromising on our ability to accurately estimate complexity due to sampling gaps.”

(11) Page 13, line 238. Since elasticity was calculated from the linear model, I wonder if the statistical significance of the model was estimated for each time series. I suppose that the elasticity value calculated for a given time series should be considered no representative if the fitting of the data to the linear model was not significant.

Thank you for this suggestion. When tested for a random subset (100 time series), the p-values for the slope of linear model (i.e. elasticity) were all extremely low ($<2.2e-16$).

There are several reasons not to explicitly include tests for significance. Given the large volume of data (hundreds of thousands of individual time series) and associated models, p-values are

likely to be problematic in measuring significance and require detailed statistical treatment. The significance value is also likely to be greatly affected by (i) the chosen sample size for measuring elasticity (currently 1000 trials) and (ii) the type of model that is used (linear vs non-linear).

We added text to our methods to highlight these questions and suggest that future studies look at this problem in detail.

Ln 346-350: “We also did not attempt to measure statistical significance, nor remove any pixels based on significance thresholds. Future studies may consider detailed statistical analyses to quantify the false discovery rate, the chosen number of threshold combinations, or the applicability of linear assumptions in the estimation of elasticity.”

(12) Page 14, line 260. Please, explain why this technique to fill gaps was used.

We add a statement explaining our choice.

Ln 361-363: “This method of resampling was chosen to ensure we had a complete time series for every pixel without adding bias.”

Figures 1 and 2. It is unclear what is the “time index” represented in the x-axis.

“Time index” refers to the day of sampling from January 1st, 1998, onwards. We have updated the figure legends.

References

- Gómez-Jakobsen, F., Ferrera, I., Yebr, L., Mercedo, J.M., 2022. Two decades of satellite surface chlorophyll a concentration (1998–2019) in the Spanish Mediterranean marine waters (Western Mediterranean Sea): Trends, phenology and eutrophication assessment. *Remt. Sensing Appl. Soc. Environ.* 28, 100855. <https://doi.org/10.1016/j.rsase.2022.100855>
- Morozov, E., Korosov, A., Pozdnyakov, D., Pettersson, L., Sychev, V., 2010. A new area-specific bio-optical algorithm for the Bay of Biscay and assessment of its potential for SeaWiFS and MODIS/Aqua data merging. *Int. J. Rem. Sens.* 31, 6541–6565. <https://doi.org/10.1080/01431161.2010.508802>.
- Staehr, S.U., Van der Zande, D., Staehr, P.A.U., Markager, S., 2022. Suitability of multisensory satellites for long-term chlorophyll assessment in coastal waters: a case study in optically-complex waters of the temperate region. *Ecol. Indicat.* 134, 108479 <https://doi.org/10.1016/j.ecolind.2021.108479>.
- Sathyendranath, S., Brewin, R.J.W., Jackson, T., M'elin, F., Platt, T., 2017. Ocean-colour products for climate-change studies: what are their ideal characteristics? *Remote Sens. Environ.* 203, 125–138. <https://doi.org/10.1016/j.rse.2017.04.017>

Reviewer #2 (Remarks to the Author):

Overview

The authors have prepared an interesting manuscript detailing the application of two statistical methods previously unused for the analysis of ocean chlorophyll. These metrics will likely be of use in future research works. The finding of potential global similarities in these metrics is also of interest. However as it stands the manuscript needs some additional work as much of the interpretation does not appear to be supported by the results; I have provided some suggestions where initial analysis may be employed below. I look forward to seeing the manuscript following review.

Major comments

The main area that should be targeted for improvement is in the interpretation of the novel application of these statistics. This particularly applies to what the statistics are revealing, as the current interpretation does not have the analysis to support it and is at times conflicting. The manuscript states these metrics measure time series complexity, however what these reflect in the real world, be it measurement error, blooms, seasonal cycling, climatic forcing or day to day variability needs to be supported by some additional analysis. I am also curious how these new metrics compare to standard deviation / variance and should be better contrasted against each other.

- Conflicting interpretations:
 - o L107 implies low elasticity is linked to stronger seasonal cycles
 - o L110-112 implies high elasticity is linked to stronger seasonal cycles
 - o Figure S3 implies elasticity is insensitive to regular seasonal cycles

We are sorry for the confusion. We have now added text to resolve these issues.

Ln 123-134: “As elasticity is sensitive to the regularity of [chl-*a*] change, regions with strong, periodic drivers of phytoplankton blooms (often manifested in seasonal or annual patterns) would naturally have lower values of elasticity in comparison to regions where the periodicity of [chl-*a*] change is irregular. Figure S2 highlights how elasticity can vary for three different model time series. If all daily changes within a time series were equal, its elasticity would equal to 0 (no sensitivity to thresholds). When evaluated for the relative frequency of daily changes, some regions were prominent in their dynamic behavior, with more days of significant [chl-*a*] increase regardless of any chosen threshold (Figure S3). Interestingly, the North Atlantic gyre had lower than average elasticity values than both Pacific gyres, despite similar values of [chl-*a*]. The difference is likely due to the stronger seasonal patterns in the North Atlantic and the influence of the annual spring bloom (40, 41), which would impart more regularity in the [chl] time series in the North Atlantic.”

- Suggestions for possible additional analysis:
 - Comparison with global seasonal amplitude
 - Comparison with global variance (/standard deviation)

Thank you for your valuable suggestions. As suggested, we conducted additional analysis to compare the results of elasticity and fractal dimension to the mean [chl-*a*], standard deviation, relative standard deviation, and the average seasonal amplitude of every time series (see Figure C below). The fractal dimension typically shows a nonlinear positive relationship with traditional measures of variability, whereas the elasticity shows a nonlinear negative relationship.

Ln 188-200: “Lastly, we also compared both complexity metrics to some traditional measures of variability for each time series (mean [chl-*a*], standard deviation, relative standard deviation, and average seasonal amplitude). The calculated fractal dimension of each time series showed a nonlinear positive relationship, whereas the elasticity showed a nonlinear negative relationship with each metric (Figures S6-7). When evaluated against each other on a similar scale, elasticity and fractal dimension did not present any strong relationships (Figures S8-9).”

◦ Application of statistics to data with seasonality removed

We did not remove seasonality for 2 broad reasons:

- (i) The seasonality and periodicity of chlorophyll time series are intrinsic components of temporal complexity.
- (ii) In some parts of the global ocean, there is an imbalance of sampling over the year (i.e. certain months of the year have little to no measurements). In an attempt to remove the seasonal cycle, we might end up propagating sampling bias.

◦ Comparison with chlorophyll product uncertainty

♣ This is available with the GlobColour product as provided by CMEMS although I’m not sure if it is temporally varying

We appreciate the suggestion. Whereas we found the idea interesting, it would likely require a deeper look into the many components of chlorophyll product uncertainty – such as the models used to merge data, the availability of *in situ* chlorophyll measurements, and the different approaches to calculating uncertainty. Such work is out of the scope of this current paper, but we have added text to our discussion citing some previous studies and the opportunity for future work.

Ln 255-261: “While we did not evaluate the chlorophyll-*a* product uncertainty in this study, future studies may consider accounting for the difference between [chl-*a*] algorithms, different uncertainty calculations, and their relationship to time series complexity (60–62). One possibility is that temporal complexity is driven by high variability within certain regions of the visible spectrum; for example, slight day-to-day differences in the green spectrum may propagate across [chl-*a*] calculations and lead to greater ‘roughness’ within time series.”

In summary sufficient additional analysis needs to be performed to (a) compare with traditional measures of variability and to (b) support the interpretation of the physical mechanism behind changes in the values of the statistics.

This paper only touches on some possible hypotheses, but it does not explicitly test the role of different mechanisms for determining temporal complexity. As such, we are agnostic in making claims about the exact sources of complexity – whether it is physical forcing, satellite error, phytoplankton growth dynamics, advection, the amount of colored dissolved organic matter (CDOM), etc. However, we performed many of the additional analysis described above to highlight relationships between time series complexity and traditional measures of variability.

Figure C: Relationship between the fractal dimension of each time series and the mean chlorophyll-*a* concentration (top left), the standard deviation (top right), the relative standard deviation (bottom left) and the seasonal amplitude (bottom right). Seasonal amplitude was calculated by subtracting the highest monthly average chlorophyll-*a* concentration from the lowest monthly average chlorophyll-*a* concentration.

Figure D: Relationship between the elasticity of each time series and the mean chlorophyll-*a* concentration (top left), the standard deviation (top right), the relative standard deviation (bottom left) and the seasonal amplitude (bottom right). Seasonal amplitude was calculated by subtracting the highest monthly average chlorophyll-*a* concentration from the lowest monthly average chlorophyll-*a* concentration.

Additionally further attempts should be made to highlight similarities and differences in the two statistical measures discussed. For example the spatial patterns in figures 1 and 2 look very similar, could these be shown on the same colour map with same standardised scale, to better highlight the similarities / differences (at least as supplementary)?

We followed the reviewer's recommendation and added a supplementary figure that uses the same standardized color scale.

Ln 198-200: "When evaluated against each other on a similar scale, elasticity and fractal dimension did not present any strong relationships (Figures S8-9)."

Figure E: Global map of elasticity (top) and fractal dimension (bottom) calculated from 25-years (2002-2022) of daily-scale chlorophyll-*a* concentration time series. Darker colors (red) indicate greater elasticity and fractal dimension for the corresponding 25x25km pixel, whereas lighter colors (grey) indicate lower elasticity and fractal dimension. Areas in white are pixels where greater than 80% of the time series were missing observations or less than 400 days with consecutive measurements. Both elasticity and fractal dimension have been min-max normalized and plotted on the same color scale.

Additionally it would be more useful if the examples of each of figure 1 and 2 covered the same grid points, including examples where the two metrics agree and also where they don't.

Elasticity and fractal dimension are independent of each other and as such, capture different aspects of the complexity of time series. We prefer to keep Figures 1 and 2 tailored to each metric. Covering the same grid points would not be informative as there is no 1:1 relationship between the two different metrics.

Instead, we added a supplementary figure plotting elasticity against fractal dimension. The new figure shows how there are no strong relationships between the two metrics.

Figure F: Relationship between the fractal dimension and elasticity of each chl-*a* concentration time series.

Finally Figures 3 (e) and (f) have vastly different colour scales with broad ranges, which may appear to show global homogeneity but may instead be an artefact of the plotting, these should be redrawn with the full range of colours for each being employed (at least as supplementary)

Thank you for this suggestion. In the revised version, we changed Figure 3 to use the full range of colors being employed, added more subdivisions, and reduced the maximum range as much as possible. This new figure reinforces our original assessment on the spatial differences of global complexity change.

Specific / minor comments

L112-114 I don't feel this statement is supported by the results as they stand, indeed no mention of climatic forcing has been made up to this point.

We have added more discussion to this section to clarify what we mean.

Ln 123-134: "As elasticity is sensitive to the regularity of [chl-*a*] change, regions with strong, periodic drivers of phytoplankton blooms (often manifested in seasonal or annual patterns) would naturally have lower values of elasticity in comparison to regions where the periodicity of [chl-*a*] change is irregular. Figure S2 highlights how elasticity can vary for three different model time series. If all daily changes within a time series were equal, its elasticity would equal to 0 (no sensitivity to thresholds). When evaluated for the relative frequency of daily changes, some regions were prominent in their dynamic behavior, with more days of significant [chl-*a*] increase regardless of any chosen threshold (Figure S3). Interestingly, the North Atlantic gyre had lower than average elasticity values than both Pacific gyres, despite similar values of [chl-*a*]. The difference is likely due to the stronger seasonal patterns in the North Atlantic and the influence of the annual spring bloom (40, 41), which would impart more regularity in the [chl] time series in the North Atlantic."

L34 / L139 If the goal of the research is to analyse outliers, outliers should probably not have been removed, as stated in L141. Alternatively a distinction needs to be made between the outliers in these different contexts.

We are sorry for the confusion. The reviewer is correct in noting that there can be outliers in different contexts.

By removing some outliers (values greater than 3 standard deviations of the mean), we are reinforcing the idea that the fractal dimension captures some fundamental property of the time series complexity in different locations. If we were to leave those values in, it becomes unclear whether the fractal dimension is entirely driven by one or two erroneous points in select locations. This is why we applied a uniform filter on a global scale. Our results still show the likelihood of outliers within a time series, but the focus is on the distribution of values rather than a few arbitrary points.

We have added text to explain our choices and distinguish between outliers. In the new manuscript, we used the term "anomalous" for the values we still consider within the analysis and outliers for the points we remove.

Ln 34-35: “These patterns can be linked to the regularity of chlorophyll-*a* concentration change and the likelihood of anomalous events within the satellite record.”

Ln 166-172: “As the box-counting method has been shown to return lower fractal dimension values in time series with sharp changes (50), we first remove most outliers from each time series (see Materials and Methods) before calculating the fractal dimension. This was done to minimize the error in our computations due to likely erroneous measurements. In this case, our global map of fractal dimension reveals the likelihood of anomalous spikes within each time series (long tails in the data), possibly due to naturally occurring dynamics, but also, atmospheric correction and ocean color algorithm error.”

L161 - Some comments should be made on the change in the time series after 2019.

We have added some comments on the change after 2019. We have also changed Figure 3 to add a gridline for the year 2019.

Ln 216-225:” When calculated annually, both the elasticity and the average fractal dimension of the global ocean had a significant shift in 2003 and a gradual return to 1998 levels thereafter (Figure 3A-C), whereas the mean [chl-*a*] showed a general decrease till 2019. Interestingly, a second transition around 2019 shows a rise in mean [chl-*a*], a sharp increase in fractal dimension and a drop in elasticity.”

177-179 - Errors may increase over time as the hardware of individual satellites age.

We change the statement to indicate “change” instead of “decrease” and added text to include the reviewer’s concern.

Ln 242-246: “Part of this change can likely be attributed to the greater number of satellite sensors in orbit (and merged in the [chl-*a*] product) and a change in the magnitude of error-related outliers. Errors may decrease due to the greater availability of data and possibility of quality control but may also increase over time as the hardware of individual satellites age.”

Figures 3 a-c – Gridlines may help identify years where changes are occurring

We appreciate the suggestion. The order has been changed accordingly.

Figure 3f - Why does this subplot have more white pixels than the subplots above it?

Thank you for mentioning this. As the elasticity calculation needs more data, there are locations where the data coverage in 2003 is insufficient to calculate elasticity. That is why there are more white pixels for the subplot indicating change. We have updated the figure legend.

Ln 291-292: “There are more white pixels in subplot (e) because of less than 100 days of observation in 2003.”

Figure 3 – The order of elasticity and fractal dimension should ideally follow the text (i.e. the plots of elasticity should appear above those of fractal dimension)

The order has been changed.

L228 - Day to day thresholds in the range of 0.01 to 0.5 % of median seem low, and I would expect changes to cross this threshold the vast majority of days, are these the correct values and if so, could the authors comment on this?

Sorry for the typo. The thresholds varied from 0.01 (which is 1%) to 0.50 (which is 50%) of the median. We have corrected this in the revised version.

Ln 318-320: “Next, we created two uniform distributions of thresholds (τ_1 and τ_2 ; 1000 values each) that ranged from 1% to 50% of the median of the individual [chl-*a*] time series.”

Reviewer #3 (Remarks to the Author):

The authors proposed new metric to quantify the complexity of Chl time series obtained from satellite ocean color remote sensing, and implied that the patterns could be explained using seasonal forcing and outlier events. This reviewer found the analyses interesting, but the results and implications are questionable.

First, the authors used daily satellite Chl products to carry out the analyses, but it is well known that daily satellite Chl is subject to many sources of uncertainties, thus the spikes in the time series (see Figs. 1 and 2) are not necessarily real oceanographic events. Note that such spikes are the main drivers of the “time series complexity” studied here. To the least, the authors should test sensitivity of the new metric on different temporal resolution and spatial resolution.

We appreciate the careful comments. Following the reviewer's comment, we re-calculated the elasticity and fractal dimension for every time series on a weekly and monthly scale. Below, we show how aggregated time series data are “smoother” than daily scale data (lower fractal dimension), but this has no critical effect on the overall spatial patterns on a global scale. Similarly, the elasticity changes, but such changes still indicate the key differences across different regions. In fact, there are greater differences that are noticed with a coarser temporal

resolution because aggregation leads to greater “regularity” in some areas. These results suggest that it is very unlikely that the complexity of time series is entirely driven by “spikes”. We discuss in the manuscript how only part of the differences across regions can be attributed to differential error rates.

We add these figures to our supplement and discuss these additional results in the revised manuscript.

Ln 180-183: “Similarly, when we tested for the influence of time series resolution (daily, weekly, monthly, etc.) on the metrics, our results showed that aggregated time series are typically ‘smoother’ (i.e., possess lower fractal dimension). The elasticity also changes with coarser resolution for some regions and indicates geographically similar, but greater differences (Figure S4).”

Figure G: Global maps of the fractal dimension (left) and elasticity (right) for a range of temporal resolutions (top row – daily scale, middle row – weekly scale, bottom row – monthly scale). Darker colors (purple) indicate greater elasticity for the corresponding 25x25km pixel, whereas lighter colors (yellow) indicate lower elasticity. Similarly, Green, and yellow indicates a higher fractal dimension for the corresponding 25x25km pixel, whereas red indicates lower fractal dimensions.

With regards to the spatial resolution, we would expect to see a similar effect because decreased spatial resolution tends to smoothen the time series. In general, smoother time series will show lower values of elasticity and fractal dimension (as above), but the overall pattern will be unchanged. As part of the goal of this study was to explicitly measure the complexity on a global scale, we chose to analyze time series at the highest resolution both temporally and spatially that was computationally feasible.

More importantly, for a specific location, the metric provides a measure of Chl variation in the past decades only, has no information on the future change or variation of Chl, or if there will be a spike on any days or periods. While complexity of the time series is useful information, it is more important to know, and understand, the temporal variation of Chl (or any other physical and biogeochemical properties) in the global ocean.

We appreciate that the reviewer recognizes that the complexity of time series is useful information. We also agree with the reviewer that it is important to understand the temporal variation of Chl in the global ocean, which is what we did in this manuscript by calculating the mean chlorophyll, elasticity, and fractal dimension across the satellite record (Fig. 3). Whereas what is more important could depend on many factors, we understand the value of predicting the future, which would require earth system modeling on a global scale. While such models have made substantial progress over the past few decades, current technology does not allow most (or likely any) global earth system models to resolve the complexity properties as shown in *in situ* data as we use in this study. It is outside the scope of this study to create an earth system model that will be able to circumvent this problem.

REVIEWERS' COMMENTS

Reviewer #1 (Remarks to the Author):

I thank the authors for considering all my comments and modifying the manuscript accordingly. I think that the manuscript has been improved strongly and I think that it is useful for publication. In my previous report, I pointed out three main concerns.

-One of them was the fact that the analyzed time series were based on a product calculated with different satellite data; my concern was that biases among satellites would affect the features of the time series obtained. The authors tested these possible biases by comparing the results of the analyses for the whole time series with time series of MODIS and found that the differences in elasticity and fractal dimension were reduced, which is fine.

-The second point was regarding to the interpretation of the elasticity based on daily changes in terms of describing the phytoplankton bloom dynamics since the bloom concept should include not only the magnitude of the chlorophyll a concentration increase beyond a threshold but also the duration of these elevated values. Additionally, as pointed out by Reviewer#3, spikes in the time series would be due to processes or inconsistencies that are not necessarily related to phytoplankton dynamics. The authors comment that it is not possible to include the bloom duration in their calculations of elasticity although they modified the text to highlight that elasticity is not a direct measurement of phytoplankton growth and that it might be influenced by other processes. Additionally, in the new version, results of elasticity and fractal dimension calculated for weekly and monthly time series are presented and a short discussion about the differences between daily and smoother time series is included.

-In my previous report, I missed some measurements of statistical deviation for the parameters of the time series. In the new version, deviations for each complexity metric have been calculated by using a suitable method.

My specific comments have been also responded satisfactorily. I have only three minor suggestions:

Lines 50-51. Satellite chlorophyll is also used to calculate primary production and then it is an important variable in the context of the analysis of the carbon biogeochemical cycle. Perhaps it should be mentioned. Additionally, I am not sure what the authors mean with “validating earth system models”.

Lines 225-227. I guess that these deviations in the spectral reflectance with respect to the global mean indicate that these areas have singular optical properties. I suggest providing some hypothesis for that (differences in the taxonomical composition of the phytoplankton, influence of terrestrial discharges?).

Line 240-242. Please, provide a short hypothesis about which these environmental or ecological drivers are.

Reviewer #2 (Remarks on code availability):

The code allows reproduction of the statistical approaches used by the authors with an appropriately explained readme file and provided test dataset. Additional coding would be required to reduce the results of the paper in their full extent.

Reviewer #1 (Remarks to the Author):

I thank the authors for considering all my comments and modifying the manuscript accordingly. I think that the manuscript has been improved strongly and I think that it is useful for publication. In my previous report, I pointed out three main concerns.

-One of them was the fact that the analyzed time series were based on a product calculated with different satellite data; my concern was that biases among satellites would affect the features of the time series obtained. The authors tested these possible biases by comparing the results of the analyses for the whole time series with time series of MODIS and found that the differences in elasticity and fractal dimension were reduced, which is fine.

-The second point was regarding to the interpretation of the elasticity based on daily changes in terms of describing the phytoplankton bloom dynamics since the bloom concept should include not only the magnitude of the chlorophyll a concentration increase beyond a threshold but also the duration of these elevated values. Additionally, as pointed out by Reviewer#3, spikes in the time series would be due to processes or inconsistencies that are not necessarily related to phytoplankton dynamics. The authors comment that it is not possible to include the bloom duration in their calculations of elasticity although they modified the text to highlight that elasticity is not a direct measurement of phytoplankton growth and that it might be influenced by other processes. Additionally, in the new version, results of elasticity and fractal dimension calculated for weekly and monthly time series are presented and a short discussion about the differences between daily and another time series is included.

-In my previous report, I missed some measurements of statistical deviation for the parameters of the time series. In the new version, deviations for each complexity metric have been calculated by using a suitable method.

My specific comments have been also responded satisfactorily. I have only three minor suggestions:

Thank you for your constructive review. We are happy to see that the reviewer recognized the improvement of the new version, enabled by reviewers' earlier comments. We respond to the suggestions below.

Lines 50-51. Satellite chlorophyll is also used to calculate primary production and then it is an important variable in the context of the analysis of the carbon biogeochemical cycle. Perhaps it would be mentioned. Additionally, I am not sure what the authors mean with "validating earth system models".

We appreciate the reviewer's comment. We have revised the text to mention primary productivity and changed "validating" to "testing".

Ln 51-53: “Chlorophyll-*a* concentration estimates are used for various goals: estimating primary productivity (3), developing ecological indicators (10), monitoring long-term trends (11), or testing earth-system-models (12, 13).”

Lines 225-227. I guess that these deviations in the spectral reflectance with respect to the global mean indicate that these areas have singular optical properties. I suggest providing some hypothesis for that (differences in the taxonomical composition of the phytoplankton, influence of terrestrial discharges?).

We incorporated the reviewer’s suggestions into the text.

Ln 234-236: “This variability could be caused by several factors – such as rapid differences in the taxonomic composition of phytoplankton or the influence of terrestrial discharge.”

Line 240-242. Please, provide a short hypothesis about which these environmental or ecological drivers are.

We provide some hypotheses of possible environmental and ecological drivers.

Ln 251-253: “Examples of possible drivers include mixed layer stability, eddy formation, temperature change and plankton growth and grazing rates.”

Reviewer #2 (Remarks on code availability):

The code allows reproduction of the statistical approaches used by the authors with an appropriately explained readme file and provided test dataset. Additional coding would be required to reduce the results of the paper in their full extent.

Thank you for checking the code. We have added an additional script that can calculate the elasticity and fractal dimension annually for the provided test dataset. We also expanded the test dataset to 5000 time series (instead of 100).

Given the extremely large file sizes required for sharing the entire satellite record on a daily scale, we recommend readers download their own data from existing repositories and reproduce the analysis in its full extent. Unfortunately, such an endeavor will also require high-performance computing and any code will need to be tailored to the reader’s specifications.